# An Oral Galectin Inhibitor in COVID-19—A Phase II Randomized Controlled Trial

**DOI:** 10.3390/vaccines11040731

**Published:** 2023-03-25

**Authors:** Alben Sigamani, Kevin H. Mayo, Michelle C. Miller, Hana Chen-Walden, Surendar Reddy, David Platt

**Affiliations:** 1Carmel Research Consultancy Pvt. Ltd., Bengaluru 560025, Karnataka, India; 2Department of Biochemistry, Molecular Biology and Biophysics, University of Minnesota, 6-155 Jackson Hall, 321 Church Street, Minneapolis, MN 55455, USA; 3Pharmalectin India Pvt. Ltd., Rangareddy 500039, Telangana, India; 4Department of Pulmonology, ESIC Medical College and Hospital, Sanath Nagar, Hyderabad 500038, Telangana, India

**Keywords:** galectin-3, ProLectin-M, SARS-CoV-2, clinical trial, spike protein

## Abstract

Background: SARS-CoV-2 vaccines play an important role in reducing disease severity, hospitalization, and death, although they failed to prevent the transmission of SARS-CoV-2 variants. Therefore, an effective inhibitor of galectin-3 (Gal-3) could be used to treat and prevent the transmission of COVID-19. ProLectin-M (PL-M), a Gal-3 antagonist, was shown to interact with Gal-3 and thereby prevent cellular entry of SARS-CoV-2 in previous studies. Aim: The present study aimed to further evaluate the therapeutic effect of PL-M tablets in 34 subjects with COVID-19. Methods: The efficacy of PL-M was evaluated in a randomized, double-blind, placebo-controlled clinical study in patients with mild to moderately severe COVID-19. Primary endpoints included changes in the absolute RT-PCR Ct values of the nucleocapsid and open reading frame (ORF) genes from baseline to days 3 and 7. The incidence of adverse events, changes in blood biochemistry, inflammatory biomarkers, and levels of antibodies against COVID-19 were also evaluated as part of the safety evaluation. Results: PL-M treatment significantly (p = 0.001) increased RT-PCR cycle counts for N and ORF genes on days 3 (Ct values 32.09 ± 2.39 and 30.69 ± 3.38, respectively) and 7 (Ct values 34.91 ± 0.39 and 34.85 ± 0.61, respectively) compared to a placebo treatment. On day 3, 14 subjects in the PL-M group had cycle counts for the N gene above the cut-off value of 29 (target cycle count 29), whereas on day 7, all subjects had cycle counts above the cut-off value. Ct values in placebo subjects were consistently less than 29, and no placebo subjects were RT-PCR-negative until day 7. Most of the symptoms disappeared completely after receiving PL-M treatment for 7 days in more patients compared to the placebo group. Conclusion: PL-M is safe and effective for clinical use in reducing viral loads and promoting rapid viral clearance in COVID-19 patients by inhibiting SARS-CoV-2 entry into cells through the inhibition of Gal-3.

## 1. Introduction

COVID-19, caused by severe acute respiratory syndrome coronavirus 2 (SARS-CoV-2), remains an unprecedented event in world history. More than 2 years after the COVID-19 outbreak, the number of infections continues to rise in many areas of the world. Despite the availability of vaccines and other approved treatments, the prolonged persistence of COVID-19 is a major concern for the economy and people’s quality of life. However, existing vaccines continue to be extremely effective in reducing disease severity and preventing hospitalization and death [1]. The continued emergence of highly transmissible or pathogenic multiple SARS-CoV-2 variants with the potential to evade existing COVID-19 vaccines and antibody therapies contributes to disease transmission and persistence [2,3]. There is still no effective standard of care for treating COVID-19 patients; thus, there is an urgent need to discover effective therapies. Potent antiviral regimens that rapidly reduce SARS-CoV-2 viral load and accelerate viral clearance in patients may hold promise in achieving the goal of preventing viral transmission and the emergence of new variants. In this context, a novel molecular approach that targets viral entry by engaging galectins is expected to be promising.

Galectins are small S-type lectins that function as pattern recognition receptors to enhance microbial invasion and regulate innate immune responses [4,5]. These selectively bind the N-acetyllactosamine (Gal1-3GlcNAc or Gal1-4GlcNAc) of N-linked and O-linked glycoproteins [6,7]. Among the 12 human galectins, galectin-3 (Gal-3) is abundantly expressed during viral infections in a variety of immune cells (neutrophils, macrophages, monocytes, and dendritic cells), as well as in fibroblasts and epithelial and endothelial cells [8]. Secreted Gal-3 not only regulates viral entry and attachment [9,10] but also mediates a number of detrimental consequences, including inflammatory responses [11]. During infections, Gal-3 was shown to cause a dysregulated pattern of pro-inflammatory cytokine production, including cytokines TNF-α, IL-1β, and IL-6, among others [12]. The role of Gal-3 in SARS-CoV-2 disease severity and associated cytokine storm syndrome was previously described [13,14].

Recognition and binding to cell receptors is a critical step in viral infection. The S1 subunit of the SARS-CoV-2 spike protein consists of an N-terminal domain (NTD) and a C-terminal domain (CTD). Both facilitate viral adherence and entry processes [15] by binding with host angiotensin converting enzyme receptor 2 (ACE2) receptors in a S1-CTD fashion [16] while S1-NTD binds with N-acetylneuraminic acid (Neu5Ac) sugar receptors, Gal-3, and sialic acid-linked GM1 ganglioside [17]. The galactose-binding domain of human Gal-3 showed a nearly identical topology to that of the S1-NTD of SARS-CoV2 [16] and also showed higher binding affinity with GM1 ganglioside [18]. It was also reported that both Neu5Ac and the carbohydrate recognition domain (CRD) of Gal-3 receptors have a COOH terminal which strongly interacts with sugar molecules, such as lactose and larger galacto-oligosaccharides, or protein molecules including SARS-CoV2 spike protein [5,17]. The abundance of Neu5Ac and Gal-3 in the human nasopharynx and oral mucosa [19], as well as their strong interaction with protein molecules, may lead to the high transmissibility and infectivity of SARS-CoV-2, particularly at viral entry points [8,20]. Thus, by understanding the numerous roles of Gal-3 in viral infection, it is believed that human Gal-3 antagonists are capable of preventing SARS-CoV2 adhesion and cell entry as well as virus-associated inflammatory responses.

The RT-PCR technique is performed to determine the viral loads in infected patients that correspond to PCR cycle thresholds (Ct values). Though it may not be relevant for clinical outcomes, it is an important measure of the SARS-CoV-2 infectivity of an individual from the threshold cycle (Ct) value [21]. Hence, any intervention shown to increase the cycle threshold, indicating a reduction in viral multiplication and, thus, infectivity, is significant. ProLectin-M (PL-M) is a novel polymeric carbohydrate derived from gaur gum that competitively binds to the NH2 terminal domain (NTD) of human Gal-3 [22]. In our previous pilot study on 10 patients with COVID-19, we found that the oral administration of PL-M (chewable tablets) increased PCR cycle threshold, indicating the elimination of SARS-CoV-2 as well as a reduction in viral load in these patients [23]. Further in vitro studies on SARS-CoV-2-infected Vero cells revealed that PL-M strongly binds to human Gal-3, resulting in a reduction in viral load in Vero cells [24]. In this study, we further share the results of a clinical trial on COVID-19 patients.

## 2. Materials and Methods

### 2.1. Study Design and Oversight

The efficacy of the study drug (Chewable Tablet Galactomannan, 1400 mg PL-M) was assessed in a randomized, double-blind, placebo-controlled clinical trial in ambulatory patients with mild to moderately severe COVID-19. The study protocol was approved by the Institutional Ethics Committee (IEC) of ESIS Medical College and Hospital, Sanath Nagar, Hyderabad, India (ESIC Registration No: ECR/1303/Inst/TG/2019). Per regulatory standards, the study was registered with the Clinical Trials Registry-India (CTRI/2022/03/040757) on 3 March 2022. The study was conducted in accordance with the Declaration of Helsinki, the Good Clinical Practice guidelines, and local regulatory requirements. Prior to any trial-related activity, written informed permission from each participant was obtained. A total of 34 participants were identified, screened, and enrolled in this study. A paper-based case report form was used to capture all clinical data.

### 2.2. Inclusion and Exclusion Criteria

The study involved 34 participants aged ≥18 years with mild to moderately severe COVID-19 who were willing to provide written informed permission and follow the trial guidelines. Participants were enrolled in the study if they had a recent rRT-PCR positive diagnosis (≤3 days) for COVID-19 with any of the following conditions: Ct value ≤ 25, hospitalization for classical (CDC-defined) COVID-19 symptoms (onset ≤ 5 days), and high-risk category of morbidity with SARS-CoV-2 infection. The exclusion criteria for participation included oxygen saturation levels (SpO2) ≤ 94% on room air, pregnant or breastfeeding women, active malignancy or chemotherapy, known allergies to any component of the study intervention, and pre-existing medical conditions that made the study protocol unsafe to follow. Participants who were receiving or had received any investigational COVID-19 treatment within 30 days before screening were also excluded from this study.

### 2.3. Study Objective and Outcome Measures

The objective of this study was to determine the efficacy of PL-M chewable tablets (1400 mg) as a galectin antagonist in patients with mild to moderately severe COVID-19. The primary endpoint was to assess the change in the absolute counts of the nucleocapsid (N) gene, open reading frame (ORF) gene, and an increase in Ct value, all of which were estimated from COVID-19 samples taken from a nasopharyngeal swab. In addition, we intended to assess the cumulative incidence of adverse events (AEs), changes in clinical biochemistry, clinical haematology, changes in blood markers of inflammation, and changes in COVID-19 antibody levels.

### 2.4. Randomization and Treatment

Patients who signed the informed consent form and met the eligibility criteria were enrolled in the study. Enrolled participants were randomly assigned to either the PL-M group or the placebo group in a 1:1 ratio. An independent biostatistician provided a computer-generated assignment randomization list and blocks with varied block sizes to the investigators. Patients were given 1 tablet (either PL-M or matching placebo, 1400 mg) every hour for 7 days, with a maximum dose of 10 tablets per day because the viral replication cycle is 8–10 h. Each participant was instructed to hold the tablet in their mouth for 1–2 min before dissolving and swallowing it. During mealtimes, such as breakfast, lunch, tea, and dinner, the subject was required to wait 30 min after the previous meal before taking the next tablet. This was done to avoid any potential reduction in blood glucose levels caused by the tablets’ ability to impede the absorption of carbohydrates taken during the meal.

### 2.5. Study Interventions

The study interventions, PL-M (derived from gaur gum galactomannan) and placebo tablets (chewable), were manufactured at the GMP facility of Murli Krishna Pharma Pvt. Ltd., in Pune, Maharashtra. All study interventions were identical in appearance, shape, colour, packaging, and texture. NMR studies showed that PL-M has a high affinity for Gal-3 (Appendix A) as well as for the native glycosylated SARS-CoV-2 spike protein (Appendix A), with lactose inhibiting the interaction.

### 2.6. Study Procedures

At all visits on days 1, 3, and 7, nasopharyngeal/oropharyngeal swab samples were taken from each patient to test for COVID-19 positivity using the RT-PCR method. Swab samples were transported to the research facility for Ct value analysis for the ORF and N protein genes. Throughout the analysis, all laboratory workers remained blind to treatment allocation. RNA extractions were performed using the QIAamp Viral RNA mini kit (#52904, Qiagen) according to the manufacturer’s standard protocol. A sample was considered negative if no Ct value was obtained and no amplification curve was observed, or if the Ct value for all three targets was >29. At each visit, clinical symptoms, adverse events (AEs), and concomitant medicines were recorded. Patient safety was assessed both at the beginning and completion of the trial (day 7). Changes in vital signs and laboratory tests such as haematology and serum biochemistry were evaluated as part of the safety assessment. Patients were monitored for 28 days from the day of randomization.

#### rRT PCR

The extracted RNA was analysed using TRUPCR^®^ SARS-CoV-2 RT qPCR KIT V-2 (#3B3043B Black Bio Biotech). In the real-time RT-PCR assay, 2 target genes, including ORF and N genes, were evaluated. Ct values obtained from a series of 5 template DNA dilutions of at least 3 separate samples were graphed on the y-axis versus the log of the dilution on the x-axis to measure the efficiency of the PCR. The Ct values assumed by the following equation were used to determine the logarithm of the recombinant gene copy numbers: Ct = slope log (Gene Copy Number) + 1, where 1 acts as the standard curve’s intercept.

### 2.7. Statistical Analysis

A total of 34 subjects were randomized, and no formal sample estimation was undertaken. Ct values and absolute copy numbers were compared using a parametric, unpaired repeated t-test with Welch’s correction or a non-parametric Mann–Whitney U test. A two-tailed significance of *p* < 0.05 was considered statistically significant.

## 3. Results

### 3.1. Baseline Characteristics

Thirty-four participants with mild to severe COVID-19 disease were screened for this study, and all met the inclusion criteria. Table 1 summarizes the baseline characteristics of the participants. Following randomization, each group had 17 participants, and all the participants had completed the study. The participants were 70.59% (24) male and 29.41% (10) female. The average age of the participants in the control group was 37.29 ± 7.73 years, while in the PL-M group it was 41.82 ± 5.27 years.

### 3.2. Assessment of Clinical Efficacy

SARS-CoV-2 RT-PCR tests were performed on days 1, 3, and 7. The results of the RT-PCR assays are presented in Table 2 and Figure 1A–C. As demonstrated in Table 2 and Figure 1A,B, the RT-PCR cycle counts for N and ORF genes in both groups increased during the treatment period (days 1–7). Mean RT-PCR cycle counts for both N and ORF genes were significantly (*p* = 0.001) higher in the PL-M treatment group in days 3 (Ct values 32.09 ± 2.39 and 30.69 ± 3.38, respectively) and 7 (Ct values 34.91 ± 0.39 and 34.85 ± 0.61, respectively). Contrarily, during the course of the study, mean RT-PCR cycle counts in the placebo group were consistently below 29 (targeted cycle count 29) for both N and ORF genes in days 3 (Ct values 24.38 ± 1.25 and 24.12 ± 1.16, respectively) and 7 (Ct values 27.24 ± 1.25 and 26.56 ± 1.22, respectively). On day 3, as can be seen in Figure 1C, RT-PCR cycle counts were above the targeted cut-off value of 29 in 14 or 82.35% of the participants in the PL-M treatment group, whereas all of the participants in the placebo group had RT-PCR cycle counts below the targeted cut-off value. Interestingly, on day 7, all of the participants in the PL-M treatment group had cycle counts above the targeted cut-off value, whereas the majority of participants (16, or 94.12%) in the placebo group had cycle counts below the targeted cut-off value.

At the baseline, the most frequent symptoms of COVID-19 in both study groups were chills, feeling feverish, cough, body aches, and pain. After 7 days, most of the symptoms completely disappeared in the PL-M group, whereas patients continued to experience chills, feverish sensations, and cough symptoms in the placebo group (Figure 2). Only 1 patient (5.88%) reported having a cough, and 1 patient (5.88%) noted a change in taste and smell. Symptom frequency also decreased in the placebo group; however, 41.18%, 29.41%, and 35.29% of patients continued to experience chills, feverish sensations, and cough symptoms after 7 days, respectively. Additional symptoms such as headache, fatigue, and body pain were observed in 5.88% of the placebo patients.

### 3.3. Assessment of Safety

The changes in vital signs were assessed at all visits, and no abnormal changes in vital signs were observed in either study group. In both groups, all haematological and serum biochemical parameters showed no abnormal changes and were within their normal ranges. During the study period, no serious adverse events were reported.

## 4. Discussion

Patients with COVID-19 can be treated with corticosteroids, a combination of different antiviral drugs, healing plasma, certain antibiotics, and supportive care [25]. However, they were not very successful in preventing disease progression [26]. Therefore, any potential pharmacological candidates that can alleviate the progression of the disease and increase resistance to COVID-19 appear promising. As stated in the introduction, virus entry into host cells is an early and critical step in viral infection, a process mediated by the S1 subunit of the spike protein in SARS-CoV2. The S1 subunit of the SARS-CoV-2 spike protein consists of an N-terminal domain (NTD) and a C-terminal domain (CTD). The main entry mechanism of SARS-CoV2 was identified as CTD binding to ACE2 receptors [27]. However, the importance of NTDs in the entry of SARS-CoV2 has been largely neglected, despite the fact that it is the main mechanism of entry in many other coronaviruses known to infect humans [28]. Furthermore, antibody-mediated neutralization of SARS-CoV2 S1-NTD was found to completely prevent virus entry into cells [29]. This evidence suggests that the NTD region is crucial for viral entry.

As previously mentioned, human Gal-3 is crucial to the entry of SARS-CoV-2, and its blockage may prevent the progression of the disease. Human Gal-3 was shown to have a significant degree of structural and sequence similarity to the NTD of SARS-CoV2 as well as to the NTD of NL63-CoV and the infectious bronchitis coronavirus [30]. This NTD was also identified to bind to Neu5Ac, which likely explains the high infectivity of SARS-CoV2, and is crucial for cell entry [17,19,31,32]. Gal-3 inhibitors that target regions of structural overlap with the NTD may have double binding capabilities, revealing a potential strategy to inhibit viral entry [33]. Recently, COVID-19 patients were shown to have higher levels of Gal-3, TNF-α, IL-1β, and IL-6 [34,35]. Gal-3 inhibition greatly reduces the levels of these cytokines, and it therefore may also hold hope for reducing the inflammatory sequelae associated with COVID-19 [34,36].

In an early in vitro study, we showed that PL-M lowered viral load in and increased its clearance from Vero cells by inhibiting the entry of SARS-CoV-2 into Vero cells following PL-M binding to the NTD of the Gal-3 molecule [24]. Furthermore, in a small clinical study of 10 COVID-19 patients, we found that PL-M treatment resulted in a quick reduction in viral load and increased viral clearance without eliciting any serious adverse effects [23]. In this randomized, placebo-controlled clinical trial (n = 34), we found that PL-M treatment significantly reduced the duration of viral clearance in COVID-19 patients, as evidenced by an increase in RT-PCR cycle counts (mean cycle counts > 29 on days 3 and 7, *p* = 0.001) for SARS-CoV-2 N and ORF genes. After 7 days of PL-M treatment, all participants were found to have higher cycle counts for both genes, which were well above the target cycle cut-off value (target cycle value 29). The placebo group had lower RT-PCR cycle counts for both genes through the end of the study (mean cycle counts < 29 on day 3). On day 7, RT-PCR cycle counts revealed that all participants in the PL-M group had cycle counts greater than 29, whereas 94.12% of participants in the placebo group had cycle counts less than 29. Additionally, most of the participants’ COVID-19 symptoms completely disappeared within 7 days. No major adverse effects or abnormalities in vital signs or haematological or biochemical parameters were identified after 7 days of PL-M administration. Our findings were consistent with those of earlier in vitro and clinical studies [23,24]. Time to recovery is an important factor contributing to the incidence of long COVID [37]. Having shorter infectivity and exposure to the virus reduces immune modulation, which is characteristic of symptoms of long-COVID [38].

In relation to Gal-3 inhibitors, it is plausible that PL-M interacts with Gal-3 in the same way as the structurally similar NTD of the S1 subunit of SARS-CoV-2. Overall, the inhibition of Gal-3 and blockade of the NTD of the S1 subunit and ACE2 receptors may be the anti-SARS-CoV-2 actions of PL-M. Gal-3 is upregulated in influenza A virus infection and supports replication. Hence, blocking Gal-3, as shown in our NMR data (Appendix A), indicates that PL-M binds specifically to Gal-3 in the µM range and could potentially create a broad-spectrum anti-influenza treatment [39].

This study has certain drawbacks. First, this was a single centre study; data from other populations and a multicentre study are required for validation. Second, the clinical effect of PL-M was evaluated with a small sample size; thus, further clinical studies with a larger sample size are required to validate these findings before PL-M can be successfully recommended in clinical practices.

## 5. Conclusions

Our findings shed light on the efficacy of PL-M in preventing SARS-CoV-2 infection and the likely mechanism of action by which PL-M prevents viral entry into cells. This study established that PL-M reduces viral load and increases viral clearance in patients with mild to severe COVID-19. PL-M has a high affinity for Gal-3 and the native glycosylated SARS-CoV-2 spike protein, with lactose inhibiting these interactions. Overall, PL-M inhibits SARS-CoV-2 entry into cells through Gal-3 inhibition. Due to its efficacy and tolerability, PL-M can potentially be used to treat and prevent COVID-19. However, larger-scale trials are required to prove the therapeutic feasibility and effectiveness of PL-M against COVID-19.

## Figures and Tables

**Figure 1 vaccines-11-00731-f001:**
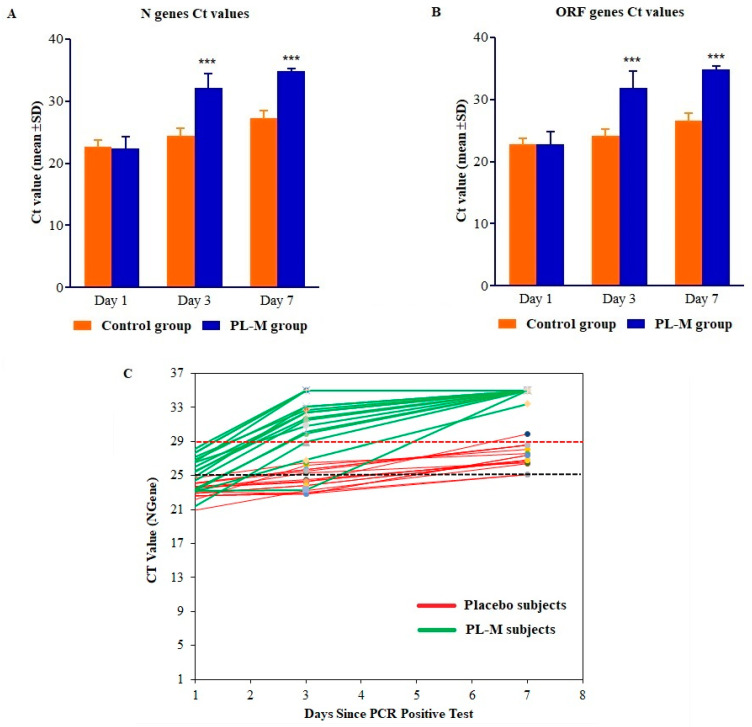
RT-PCR profiles of (**A**) the N gene, (**B**) the ORF gene, and (**C**) change in Ct value of individual subjects in the placebo and PL-M groups. RT-PCR profiles of N and ORF genes detected SARS-CoV-2 in COVID-19 patients on days 1, 3, and 7. About 82.35% of patients in the PL-M group (n = 17) became RT-PCR-negative for N and ORF genes (cycle counts > 29) on day 3 and all patients by day 7. All patients in the placebo group (n = 17) remained RT-PCR-positive for N and ORF genes on days 3 and 7 (cycle counts < 29). Changes in RT-PCR cycle counts for N and ORF genes in the PL-M group at days 3 and 7 were statistically significant compared to the placebo group (*p* = 0.001) ***.

**Figure 2 vaccines-11-00731-f002:**
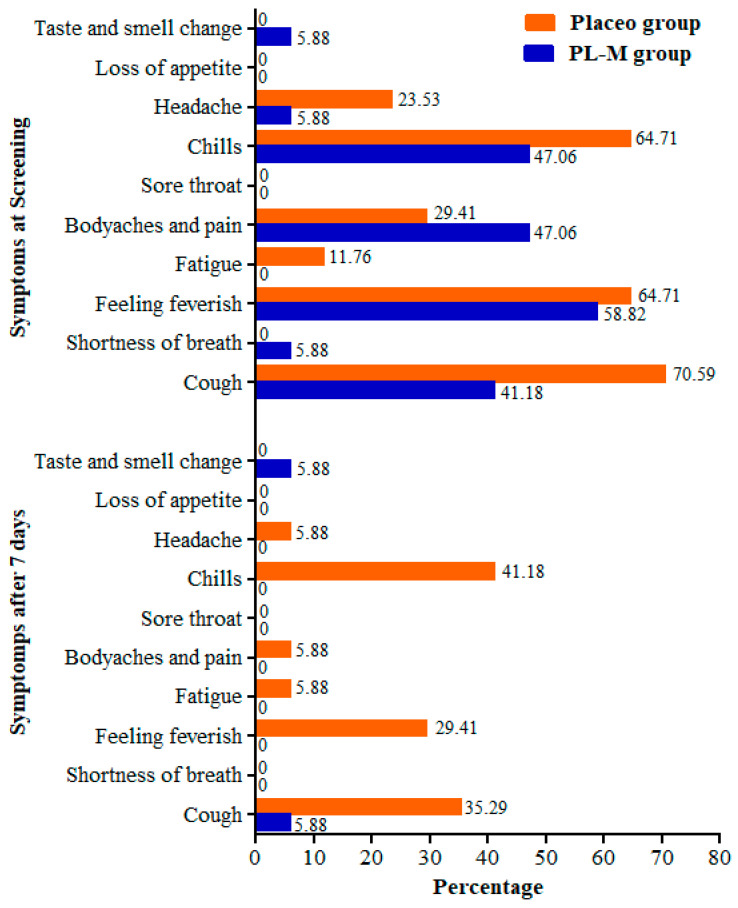
Frequency of COVID-19 symptoms in the study groups before and after treatment.

**Table 1 vaccines-11-00731-t001:** Baseline data.

Characteristics	Control Group (n = 17)	PL-M Group (n = 17)	Total
Age	37.29 ± 7.73	41.82 ± 5.27	34 (100%)
Male, n (%)	12 (70.59%)	12 (70.59%)	24 (70.59%)
Female, n (%)	5 (29.41%)	5 (29.41%)	10 (29.41%)
rRT PCR confirmed SARS-CoV-2 infection	17	17	34
N-gene cycle counts, mean, SD	22.60 ± 1.08	21.93 ± 1.88	
ORF-gene cycle counts, mean, SD	24.12 ± 1.16	29.68 ± 3.38	
Co-existing conditions (Co-morbidity)	Nil	Nil	

**Table 2 vaccines-11-00731-t002:** RT-PCR cycle counts for N and ORF gene at different visits.

COVID Tests	RT-PCR Cycle Counts (Ct Values)		*p*-Value
N Gene	Control Group (n = 17)	PL-M Group (n = 17)	
Day 1	22.60 ± 1.08	22.32 ± 2.0	0.288
Day 3	24.38 ± 1.25	32.09 ± 2.39	0.001
Day 7	27.24 ± 1.25	34.91 ± 0.39	0.001
**ORF Gene**			
Day 1	22.80 ± 1.0	22.39 ± 2.11	0.393
Day 3	24.12 ± 1.16	30.69 ± 3.38	0.001
Day 7	26.56 ± 1.22	34.85 ± 0.61	0.001

## Data Availability

All data supporting this study are provided in the main article and in supplementary information files, and they may also be provided upon request to the corresponding author.

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
