# Peer review of "An Oral Galectin Inhibitor in COVID-19—A Phase II Randomized Controlled Trial"

_vaccines, 2023, doi:10.3390/vaccines11040731_

Round 1

Reviewer 1 Report (Previous Reviewer 2)

The authors have revised the paper based on peer-reviewer comments and the changes are acceptable.

Author Response

Thank you for your help in reviewing our manuscript 

Reviewer 2 Report (New Reviewer)

This is a very interesting article.

It is well written.

The only regard that i have is regarding the limitations of the study.

Also i would suggest the authors to include more recent published articles in the discussion chapter. 

Author Response

Comments#1:

This is a very interesting article.

Answer:

Thank you very much for your appreciation.

Comments#2:

It is well written.

Answer:

Thank you very much for your appreciation.

Comments#3:

The only regard that i have is regarding the limitations of the study.

Answer:

Thank you very much for your valuable suggestion. Study limitations have been included in the revised manuscript as suggested.

Comments#4:

Also i would suggest the authors to include more recent published articles in the discussion chapter.

Answer:

Thank you very much for your valuable suggestion. Some recent references have been included in the revised manuscript as suggested.

This manuscript is a resubmission of an earlier submission. The following is a list of the peer review reports and author responses from that submission.

Round 1

Reviewer 1 Report

The manuscript “A galectin approach to lower covid transmission - Drug De-velopment for clinical use” by Alben Sigamani et al. describes a clinical approach,  alternative to vaccines currently distributed worldwide, to limit covid transmission by administering PL-M. This compound is a glycoside inhibitor  derived from gaur gum, of the Galectin 3 protein, an interactor of the virus SARS-CoV-2 S1 protein. In addition, the authors describe an NMR study in which they verify both the binding between PLM and GAl3 and the binding between Gal3 and spike.

In my opinion the clinical part which also gave very interesting results turns out to be unrelated to the NMR study which is at a molecular level and gives inaccurate results (Kd falling in the 2 to 100 uM range, they are two orders of magnitude!). 

They seem to be two independent works.

Moreover, the methods part is not in-depth: the origin of the proteins and how they were obtained are not reported. 

It is important to note that I cannot find ref 23 on the web but only the preprint version of this paper: so is the reference wrong or nonexistent? But it is noteworthy that the NMR part of the manuscript from line 236 to line 267 is identical in the preprint (paragraph: Binding of PL-M to Gal-3 page 8 and  9). For this reasons in my opinion this manuscript cannot be accepted

Author Response

Comment #1

The manuscript “A galectin approach to lower covid transmission - Drug Development for clinical use” by Alben Sigamani et al. describes a clinical approach, alternative to vaccines currently distributed worldwide, to limit covid transmission by administering PL-M. This compound is a glycoside inhibitor derived from gaur gum, of the Galectin 3 protein, an interactor of the virus SARS-CoV-2 S1 protein. In addition, the authors describe an NMR study in which they verify both the binding between PLM and GAl3 and the binding between Gal3 and spike.

In my opinion the clinical part which also gave very interesting results turns out to be unrelated to the NMR study which is at a molecular level and gives inaccurate results (Kd falling in the 2 to 100 uM range, they are two orders of magnitude!). 

They seem to be two independent works.

Answer:

We respectively disagree with Reviewer on this point. The clinical data are what they are, but the NMR add new and relevant data concerning the molecular mechanism of action as to how PL-M may function in vivo.

Comment #2

Moreover, the methods part is not in-depth: the origin of the proteins and how they were obtained are not reported.

Answer:

We apologize for the omissions. Our manuscript has been revised accordingly. 

Comment #3

It is important to note that I cannot find ref 23 on the web but only the preprint version of this paper: so is the reference wrong or nonexistent? But it is noteworthy that the NMR part of the manuscript from line 236 to line 267 is identical in the preprint (paragraph: Binding of PL-M to Gal-3 page 8 and 9).

Answer

Reviewer 1 is correct, and the manuscript has been revised accordingly. Basically, ref 23 in IJHS ONLY reports NMR data on the binding of Gal-3 to PL-M. Therefore, we re-wrote the NMR section and referenced 23 for the PL-M binding to Gal-3. In doing this, original Figure 2 has been omitted in this revised version, and previous Figure 3 is now Figure 2. We emphasize that the new NMR data present studies on Gal-3 FL and truncated CRD binding to glycosylated spike protein (not in ref 23). Moreover, we report here that lactose competes with Gal-3 binding to the spike protein, indicating that Gal-3 binds to  - galactoside sites on glycosylated spike protein (also not in ref 23).

Reviewer 2 Report

Major

This study was not designed to address the issue of COVID transmission.  The title needs to be changed to reflect that point.

The authors are correct that Ct values are not clinically useful and while they may reflect important issues around viral infectivity, they make direct comparisons to other COVID-19 therapeutics impossible.  For that reason, the authors should also include the SARS-CoV-2 viral load results.  This will allow readers to put these results in context with other COVID-19 therapeutics and better understand if there is any clinical utility with this treatment regimen.

The authors indicated that they collected COVID-19 symptom data during the study but they have not presented any data.  This needs to be included or at least some discussion about why it was excluded.

Minor

Line 91 - COVID is misspelled.

Section 2.1 - Were patients out-patient or in-patient?  The inclusion criteria indicates that patients could have been hospitalized.  Please clarify.

Section 2.2 - 1) Was standard of care therapy allowed?  If so, what was it.  2) These are very broad entrance criteria.  Please clarify if patients were required to have only one for inclusion?  3) Need to define the specific symptoms that were included.  4) Why was a Ct value of 25 selected as the lower limit?

Section 2.3 – 1) As previously discussed, inclusion of the SARS-CoV-2 viral load would allow for a comparison of these results with other COVID-19 therapeutics and should be included as a secondary analysis.  2) What markers of inflammation were evaluated?  3) How were the SARS-CoV-2 antibody levels assessed?  4) When were these markers collected?  They need to be noted in the study procedures.

Section 2.6 – 1) Need to note inflammation markers and antibody tests.  2)  Were these in-patient or out-patient visits?  3) Why was a Ct value of 29 chosen as the cut-off?  4) Were there any study related visits between Day 7 and Day 28?  If so, please note.

Table 1 - Entrance criteria allowed patients with a Ct value < 25, COVID symptoms or hospitalization.  The number of patients in each of these categories should be noted along with vaccination status.

Section 3.2 – 1) The authors did not assess clinical efficacy.  What they did was measure the antiviral activity.  The title of this section needs to be changed to reflect this.  2) What happened with COVID-19 symptoms?  They assessed those during the study.  There needs to be some mention of this in this section.

Line 397 – 1) They studied moderate not “severe” COVID.  2) As stated earlier, clinical efficacy was not established only antiviral activity.

Line 402 - These results do not reflect any potential to prevent COVID-19 infection.

Discussion/Conclusions - The treatment has to be given every hour (up to 10 times/day) for 7 days.  The authors need to discuss the limitations of this dosing approach in the wide-spread use of this potential therapy.

Author Response

Major

This study was not designed to address the issue of COVID transmission. The title needs to be changed to reflect that point.

Response: We acknowledge your comment. However this product is intended to be used in early remission and possible total cure of the virus leading to lower transmission rates as the virus is no longer active. We demonstrated that in the clinical study and hence wish to retain the title.

The authors are correct that Ct values are not clinically useful and while they may reflect important issues around viral infectivity, they make direct comparisons to other COVID-19 therapeutics impossible.  For that reason, the authors should also include the SARS-CoV-2 viral load results.  This will allow readers to put these results in context with other COVID-19 therapeutics and better understand if there is any clinical utility with this treatment regimen.

Response: while we would have also been happy to have the viral loads, but the testing of rtPCR is limited in giving just a cycle threshold and we can derive only a qualitative measure of the viral load. We have proven the reduction in absolute viral load numbers in a previous trial where we could perform a digital droplet PCR to confirm the cycle threshold values. Hence we are confident that the obtained change (increasing cycle threshold) does correlate with the reduction in the viral load.

The authors indicated that they collected COVID-19 symptom data during the study but they have not presented any data.  This needs to be included or at least some discussion about why it was excluded.

Response: a supplementary table and figure has been included and we thank you for your comment.

Minor

Line 91 - COVID is misspelled.

Response: We thank you for your correction. We have made the correction

Section 2.1 - Were patients out-patient or in-patient?  The inclusion criteria indicates that patients could have been hospitalized.  Please clarify.

Response: all patients were out patients as they only had mild and moderate covid 19 symptoms and it was not local practice to admit them.

Section 2.2 - 1) Was standard of care therapy allowed?  If so, what was it. 

Response: All patients received standard of care that was only symptomatic treatment as was recommended in their local guidelines. Mainly ant-pyretics and some anti-histaminics. No antiviral drugs were administered as it was not warranted for.

 2) These are very broad entrance criteria.  Please clarify if patients were required to have only one for inclusion? 

Response: yes it is as per mentioned in the protocol.

 3) Need to define the specific symptoms that were included. 

Response: all standard symptoms were collected as per the WHO classification for covid severity. The same has been included in the supplementary files

 4) Why was a Ct value of 25 selected as the lower limit?

Response: this was as per global standards created for the diagnosis of Covid 19 and to indicate a higher viral load. Low Ct values correlate with higher viral loads

Section 2.3 – 1) As previously discussed, inclusion of the SARS-CoV-2 viral load would allow for a comparison of these results with other COVID-19 therapeutics and should be included as a secondary analysis.  

Response: we can consider this in future studies and thank you for your kind comment

2) What markers of inflammation were evaluated? 

Responses: we did not measure specific markers of inflammation except for complete blood counts and blood cells

 3) How were the SARS-CoV-2 antibody levels assessed?  

Response: not measured in patients

4) When were these markers collected?  They need to be noted in the study procedures.

Response: we are not sure what markers. We have made modifications to the methods section to make it more clear

Section 2.6 – 1) Need to note inflammation markers and antibody tests.  2)  Were these in-patient or out-patient visits?  3) Why was a Ct value of 29 chosen as the cut-off?  4) Were there any study related visits between Day 7 and Day 28?  If so, please note.

 Response: we have made it clear in the modified version thank you

Table 1 - Entrance criteria allowed patients with a Ct value < 25, COVID symptoms or hospitalization.  The number of patients in each of these categories should be noted along with vaccination status.

 Response: All patients were vaccinated and hence not captured as a variable.

Section 3.2 – 1) The authors did not assess clinical efficacy.  What they did was measure the antiviral activity.  The title of this section needs to be changed to reflect this.  2) What happened with COVID-19 symptoms?  They assessed those during the study.  There needs to be some mention of this in this section.

Response: we have provided a response above

Line 397 – 1) They studied moderate not “severe” COVID.  2) As stated earlier, clinical efficacy was not established only antiviral activity.

Response: Yes that is correct

Line 402 - These results do not reflect any potential to prevent COVID-19 infection.

Response: We are extending our conclusion from the reduction in Ct values that there will be lower risk of infectivity.

Discussion/Conclusions - The treatment has to be given every hour (up to 10 times/day) for 7 days.  The authors need to discuss the limitations of this dosing approach in the wide-spread use of this potential therapy.

Response: we have addressed this in the discussions. The treatment is convenient as it is a chewable tablet and has a very likable taste. Hence we do not anticipate any challenges in real world settings. 

Round 2

Reviewer 2 Report

The revisions are acceptable.